# Multicountry genomic analysis underscores regional cholera spread in Africa

Cholera remains a significant public health burden in many countries in sub-Saharan Africa, though the exact mechanisms of bacterial emergence and spread remain largely undefined. We generated genomic data from 763 *Vibrio cholerae* O1 isolates predominantly collected between 2019-2024 to create the largest dataset of *V. cholerae* genomes sequenced locally in Africa. This dataset enabled us to interrogate recent patterns of spread, including the rapid circulation of the AFR15 lineage associated with unusually large outbreaks in Southern Africa. We provide evidence for the movement of the AFR15 lineage into new African Union Member States and confirm previously observed differences in *V. cholerae* transmission dynamics in West versus East Africa, though cross-border transmission is prevalent on both sides of the continent. Despite observed differences, evolutionary processes are similar across lineages and we find no evidence for significant changes in antimicrobial resistance genotypes. Overall, our findings emphasize the importance of regionally coordinated cross-border surveillance and interventions, while also demonstrating the critical role of locally generated genomic data in understanding the spread of cholera in Africa.

Cholera, a diarrheal disease caused by the bacterium *Vibrio cholerae*, poses a significant public health threat globally. There have been seven major global outbreaks (pandemics) of cholera, with the current outbreak primarily affecting and persisting in Africa since its introduction from Asia in 1970[1,2]. Seventh pandemic cholera is typically caused by the *V. cholerae* O1 serogroup and El Tor biotype (referred to as 7PET *V. cholerae*), and several prior studies have documented the intermittent and often seasonally driven outbreaks observed in many African countries[3–5]. These recurring outbreaks underscore the ongoing challenges in controlling and preventing cholera, particularly in resource-limited settings, and emphasize the need for comprehensive control strategies that use all available tools to better understand the epidemiology and dynamics of cholera transmission.

Severe, protracted, and persistent cholera outbreaks in several countries necessitate improved cholera surveillance and control[6]. Over 92,000 cholera cases were reported across 16 African countries from January to May 2024[7], with Zambia (20,113 cases), the Democratic Republic of the Congo (DRC; 16,539 cases), and Mozambique (7,762 cases) among the most affected[8]. Mozambique and Malawi also

experienced severe outbreaks in 2022[9], with the outbreak in Malawi recorded as one of the largest (over 56,000 cases) and deadliest (>3% case fatality rate) outbreaks in history[10,11]. Extreme weather events and the COVID-19 pandemic, with its disruptions to healthcare systems and changes in societal behaviors, likely contributed to the high morbidity and mortality of these outbreaks. However, simultaneous outbreaks in neighboring countries have been previously observed in this region and across Africa, suggesting that cholera transmission is regional and crosses borders, which has critical implications for its control and elimination[12,13].

Investigating transmission dynamics and unusual disease presentation (for example, high case fatality rates) requires an in-depth look at not only epidemiological and clinical data, but at the *V. cholerae* bacterium itself. Whole genome sequencing provides information about the specific strain(s) circulating, which can determine whether outbreaks in neighboring countries are connected, or if recent changes in the bacterial genome may explain observed trends. As a result, genomic analysis has recently emerged as an important tool in understanding the spread of cholera in sub-Saharan Africa[2]. Using

✉ e-mail: swohl@bwh.harvard.edu; SofoniasT@africacdc.org

whole genome sequences generated from *V. cholerae* isolates around the continent, previous studies first identified at least three waves of global transmission from Asia to Africa during the seventh pandemic[14], which was later divided into at least 17 independent introductions of 7PET into Africa, deemed the AFR1-AFR17 (or equivalently known as T1-T17) lineages[2,10,15–17]. These lineages have since been used to connect recent African and Middle Eastern outbreaks[16], to demonstrate regional transmission patterns in both East[17–19] and West Africa[13], and to suggest a possible transmission route underlying recent cases[10].

Although genomic analyses have been effectively used to better understand cholera transmission on the African continent, previous work has also highlighted the gaps in our current understanding and the need for additional surveillance and analyses. In particular, most published work focuses on outbreaks prior to 2020, when genomic surveillance infrastructure in Africa was limited. Therefore, an updated picture of *V. cholerae* diversity is needed, including how transmission patterns have or have not changed in recent years, the role of environmental reservoirs in transmission, and whether changes in the *V. cholerae* bacterium may have contributed to the worst cholera outbreaks in decades. There are also growing concerns over the emergence of antibiotic resistance to drugs used to manage severe cases of cholera and other infectious diseases[20], especially because bacteria can share resistance genes via horizontal gene transfer[21]. Finally, questions remain about the ability of genomic data to provide information that can meaningfully be used to inform and prioritize intervention approaches such as organizing vaccination campaigns.

In response to these unanswered questions, the Africa Centres for Disease Control and Prevention (Africa CDC), through the Africa Pathogen Genomic Initiative, formed the Cholera Genomics Consortium in Africa ("CholGEN") in collaboration with national public health institutes and national reference laboratories from seven African Union (**AU**) Member States (Cameroon, DRC, Malawi, Mozambique, Nigeria, Uganda, and Zambia) and international partners[22]. These Member States were selected based on public health needs, represent distinct cholera transmission scenarios (primarily endemic versus primarily caused by imports), and have collectively observed most known African lineages. Additionally, several of these countries share borders, which will aid in understanding the role of cross-border transmission. As part of CholGEN, laboratorians and bioinformaticians from each of these member states have been working to leverage local genomic sequencing capacity built during the COVID-19 pandemic to sequence bacterial isolates from recent cholera outbreaks. The resulting data provide an updated picture of circulating cholera diversity, point to specific instances of likely cross-border transmission, and highlight the role of genomic data in understanding different transmission scenarios. These results, made possible by a highly collaborative effort, underscore the importance of multicountry solutions to cholera control, which should include coordinated genomic surveillance that can reveal both cross-border transmission events and notable changes in the bacterial genome.

## Results

Through CholGEN, the seven AU Member States identified, processed, and sequenced 1220 isolates in-country (Table 1, Fig. 1A), resulting in a final dataset of 763 high-quality genomes included in downstream analyses (Supplementary Data 1; see Methods for exclusion criteria). Each participating Member State selected cholera samples for sequencing based on geographical distribution and completeness of metadata, to obtain a representative picture of national and regional cholera spread. The isolates sequenced by CholGEN represent the vast majority of cholera genomes sequenced in Africa since 2019 and the largest dataset of *V. cholerae* genomes sequenced locally in Africa (Fig. 1B). This density of recent data enabled us to investigate the persistence of circulating strains, recent changes in the cholera genome, and the frequency of cross-border transmission.

To assess whether the most recent cholera outbreaks were derived from the known diversity of *V. cholerae* in Africa or novel introduction events, we performed a Bayesian phylogenetic reconstruction of the third wave of 7PET, which includes all lineages known to be currently circulating in sub-Saharan Africa (AFR9-AFR17[2]). The reconstruction included 739 high-quality genomes in this dataset and 1764 publicly available whole genome sequences from Africa and elsewhere (Fig. 2A, Supplementary Data 2). We assigned newly generated sequences to lineages using their phylogenetic placement and found that they all descended from previously described introductory events (Fig. 2B). Consistent with prior analyses, our data indicated that different regions of Africa have distinct transmission patterns. In Western and Central African Member States—Cameroon, DRC, and Nigeria—our data showed that lineages AFR10 (DRC) and AFR12 (Nigeria, Cameroon) were still circulating, consistent with previous reports[13,23]. This pushes the persistence of AFR10 and AFR12 in the region to at least 29 and 15 years, respectively. Our data also supported previous studies showing that Eastern and Southern Africa continue to maintain a greater diversity of circulating lineages[17,24,25]. We observed AFR10, AFR11, AFR13, and AFR15 circulating in the region, with two or more lineages often present in a country over a five-year period.

Many of the newly generated genomes were classified as AFR15 and may therefore provide updated insight into the emergence of this lineage, which was previously linked to the unusually large outbreak in Malawi and major outbreaks in the Middle East, South Africa, and Zimbabwe[10,26]. In addition to these locations, we identified AFR15 isolates in DRC, Mozambique, and Zambia, indicating that AFR15 continues to spread rapidly across Southern Africa and has now been introduced into Central Africa (Fig. 2B). The newly identified presence of this lineage in DRC is likely linked to the outbreak in Zambia, since all AFR15 sequences from DRC were collected in the province of Haut Katanga, a province on the Zambian border. We also observed cholera transmission between these two countries elsewhere in the phylogeny (AFR10), suggesting that movement across national boundaries plays an important role in the maintenance of cholera in this region[27].

Despite the likely importance of cross-border transmission, our phylogenetic analysis also suggested that the AFR15 lineage may represent multiple introductions into Africa from Asia, rather than a single introduction identified by prior investigations[10,26,28] (Fig. 2A). Our analysis showed that most AFR15 isolates descended from an introduction from Southern Asia in early-2020 (95% highest posterior density [**HPD**]: June 2019 to August 2020), while a smaller subset of 11 isolates originated from an introduction from the Middle East in mid-2022 (95% HPD: April 2022 to January 2023; inset of Fig. 2A), coinciding with outbreaks in Iraq, Lebanon, Syria, and Pakistan[26,28–30]. Isolates

**Table 1 | Number of *V. cholerae* genomes collected and analyzed from each of the CholGEN AU Member States**

|  | Total sequences | High quality | Included in analysis |
|---|---|---|---|
| Cameroon | 94 | 71 | 71 |
| DRC | 216 | 155 | 153 |
| Malawi | 136 | 70 | 69 |
| Mozambique | 108 | 49 | 33 |
| Nigeria | 293 | 132 | 131 |
| Uganda | 73 | 49 | 47 |
| Zambia | 300 | 237 | 235 |
| *Total* | *1220* | *763* | *739* |

High quality represents sequences which had >90% reads mapped to the *V. cholerae* O1 reference and <10% ambiguous nucleotides. "Included in analysis" refers to high-quality sequences that had a collection date available and were not excluded by clock-rate or phylogenetic placement filters (see Supplementary Data 1).

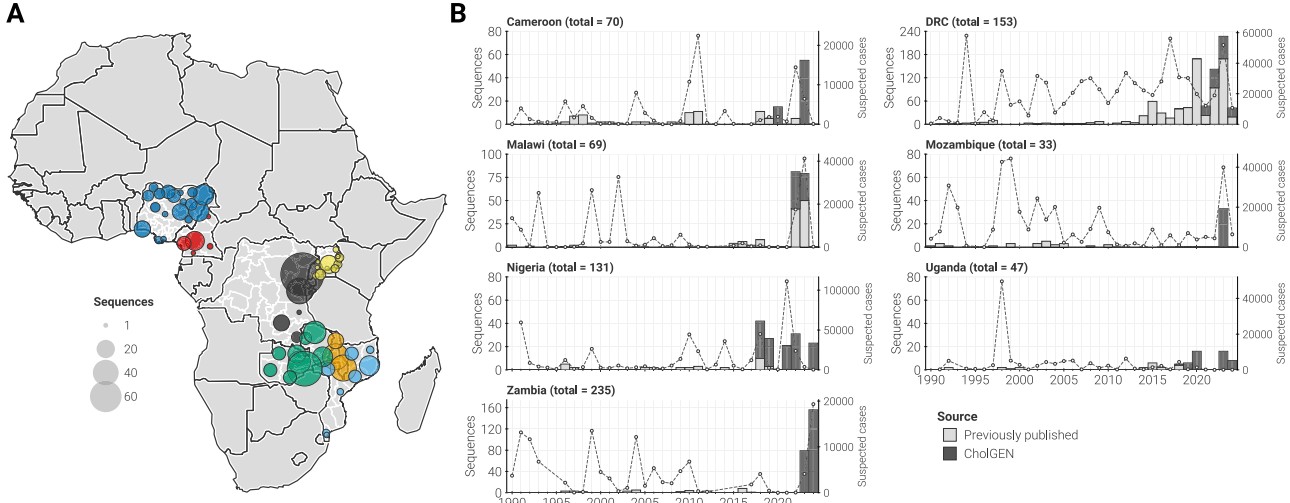

**Fig. 1 | Genomic surveillance of *Vibrio cholerae* in seven AU Member States.** **A** Map of Africa showing the distribution of sampling location for genomes generated. Points, located at the first-level administrative division, are colored by country of collection and sized by the number of high-quality genomes generated from that specific location. **B** Genomes generated per country and per year (left axis). Light gray bars: number of genomes previously published. Dark gray bars: number of genomes generated by this study. Secondary (right) axis: annual number of cholera cases reported to the World Health Organization.

from all six AU Member States associated with the AFR15 lineages were linked to the earlier and larger introduction, while only Mozambique and Zimbabwe isolates descended from the later introduction. Generation and analysis of additional sequences from these AFR15 subclusters will reveal if major outbreaks can be attributed to one or multiple introductions.

## Limited genotypic differences between lineages

We observed that the ancestral branch leading to the AFR15 lineage was long, not just in terms of time but also in terms of genetic changes (Fig. 2A & Supplementary Fig. 1), calling into question whether large mutational changes or altered substitution rates were associated with the emergence of new strains in Africa (Fig. 2A). To determine whether high rates of mutation accumulation accompanied the emergence of any of the lineages, we assessed whether any lineages were more divergent than expected given their sampling dates, and found that genomes from lineages observed to be currently circulating in CholGEN Member States (AFR10, AFR11, AFR12, AFR13, and AFR15) did not deviate significantly from the expected number of substitutions (Supplementary Fig. 2). We also estimated the mean substitution rate for each lineage using an uncorrelated relaxed clock rate model (see Methods), and found that all lineages share a similar substitution rate distribution (Supplementary Fig. 3). However, changes in transmission dynamics, host relationships, and interventions have been shown to affect the evolutionary pressure without altering the overall substitution rate of the pathogen[31–33]. Therefore, we considered other signals of selection. Specifically, we compared mutation profiles across lineages and found no significant differences in specific mutation types or nonsynonymous to synonymous mutation ratios (Supplementary Fig. 4). Gene set enrichment analysis using the biological function of *V. cholerae* genes also failed to discriminate any lineages from a null model wherein mutations were randomly distributed across the genome (Supplementary Fig. 4). Therefore, we concluded that evolutionary processes are similar across lineages despite differences in where they are found, and that other factors must be responsible for epidemiological differences, such as the rapid spread of the AFR15 lineage.

The spread of antibiotic resistance may also be associated with changes in the epidemiological patterns of bacterial spread[34] and is essential to informing best practices for cholera treatment when outbreaks occur. To provide an updated picture of the spread of resistance, we examined all isolate genomes for the presence of AMR genes and found that the AMR profiles of isolates from most CholGEN Member States did not change throughout our sampling period (Fig. 2C), even when we looked at a fine temporal resolution (Supplementary Fig. 5). Certain resistance-associated genes, particularly those that are canonically present in the *V. cholerae* core genome (*almE*, *almF*, *almG*), were present in all isolates from all countries. Other genes were associated with the lineage circulating in a region, including *floR*, *sul2*, and aminoglycoside-phosphotransferase genes that are all absent in AFR12 and AFR13 strains[35]. We did not observe any genes associated with resistance to tetracyclines, which is the primary class of antibiotics used for the treatment of cholera[36]. However, several countries observed phenotypic changes in resistance during the time period associated with these samples[21,37], suggesting the need for additional studies comparing genotypic and phenotypic AMR results[38], as well as continued monitoring of resistance in affected regions.

We found a single exception to the above in our dataset. We observed that isolates from Uganda gained several AMR genes from 2020 to 2023, including *aad2*, *blaPER-7*, *mph(A)*, *mph(E)*, *mrs(E)*, and *sul1* (Fig. 2C). This was not due to the introduction of a new lineage because only AFR13 was observed in Uganda during this period (Fig. 2B). AMR in cholera is often associated with the acquisition of plasmids containing multidrug resistance elements, so we cross-referenced the AMR genes to determine if they were present in the *V. cholerae* genome or on mobile genetic elements. We found that the AMR genes acquired by the isolates in Uganda by 2023 were not on the core genome, and were all found on an IncA/C plasmid (pCNRVC190243), which is known to be a primary source of multidrug resistance for cholera[39,40]. This plasmid has been found sporadically in historical 7PET isolates (Supplementary Fig. 6) and was not found in isolates from Uganda collected in 2018–2019, though it was observed in isolates from the 2018–2019 AFR13 outbreak in Yemen and in three Lebanese isolates that phylogenetically clustered with the 2023 isolates from Uganda[28,41]. Additionally, the plasmid was observed in European travellers returning from Eastern Africa[42,43], and a similar IncA/C plasmid containing a modified antimicrobial resistance repertoire was observed in Zimbabwe in 2018[44], suggesting that such acquisitions are not uncommon. Phylogenetic placement of 2023 sequences from Uganda containing the plasmid (which were collected from districts along the shore of Lake Victoria) suggest a recent acquisition (Fig. 2A).

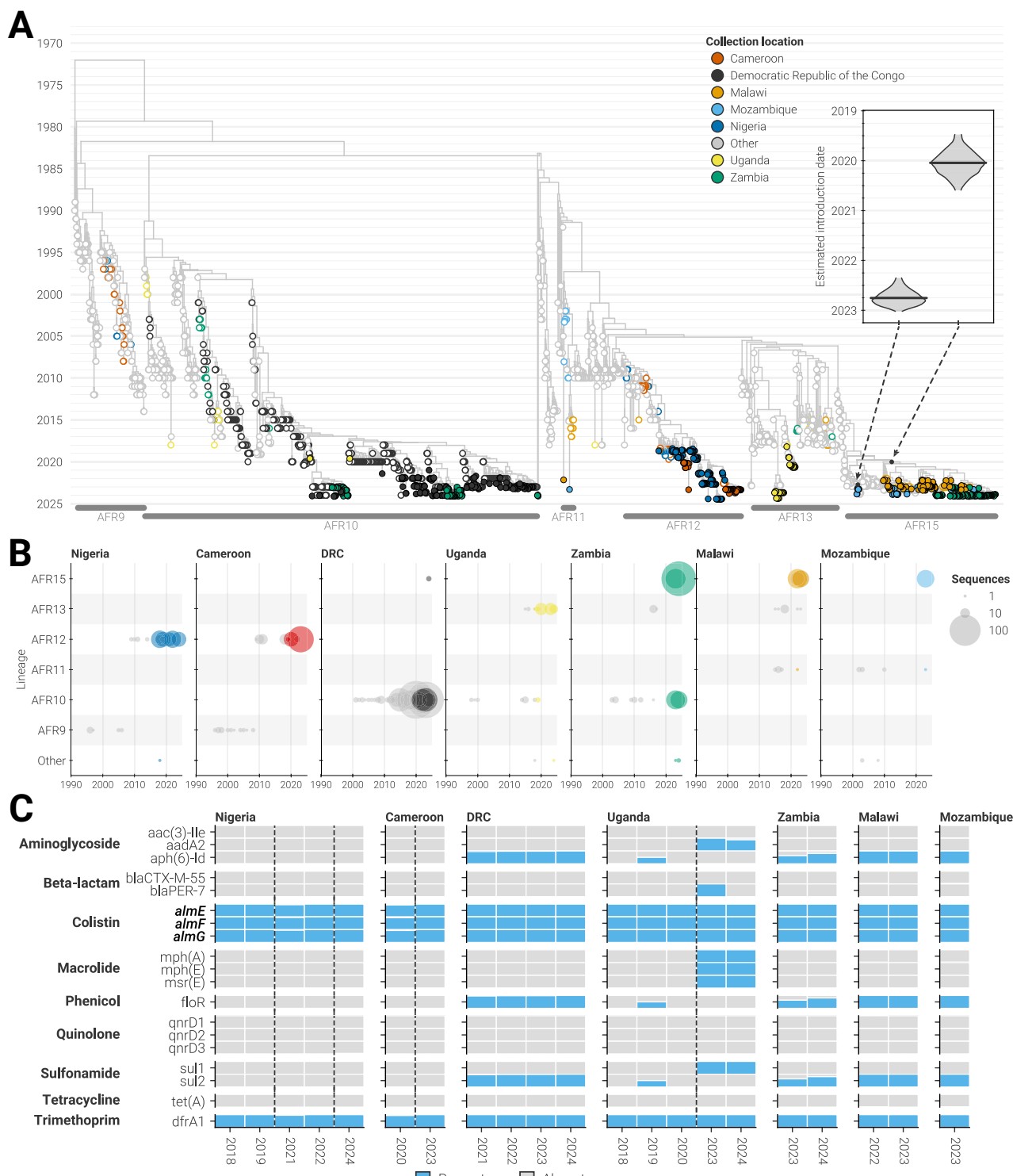

**Fig. 2 | Phylogenetic diversity of cholera. A** Maximum clade credibility tree of the third wave of 7PET *V. cholerae*. Taxa and internal nodes are colored according to their collection locations. Filled circles: genomes generated by CholGEN Member States. Open circles: previously published genomes. Black dots on nodes indicate the two putative introductions of AFR15 into Africa. Inset shows the posterior estimated date of each putative introduction. Lineages with less than 20 sequences are not labeled. **B** The annual number of genomes generated from each of the CholGEN Member States according to which inferred lineage they descend from. Size of circle scales is based on the number of high-quality genomes generated.

Color of circle describes whether the genomes were previously published (gray) or generated by this study (other colors). Countries are ordered left-to-right based on their geographic location West-to-East. Sporadic lineages with less than 20 sequences are combined as "Other." **C** The proportion of isolates collected in each year and country that carry specific antimicrobial resistance genes. Blue bars: gene present. Gray bars: gene not observed. Countries are ordered as in **B**, and genes are ordered by the class of antibiotic that they provide resistance to. Bolded genes: genes located on either genomic chromosome of *V. cholerae*.

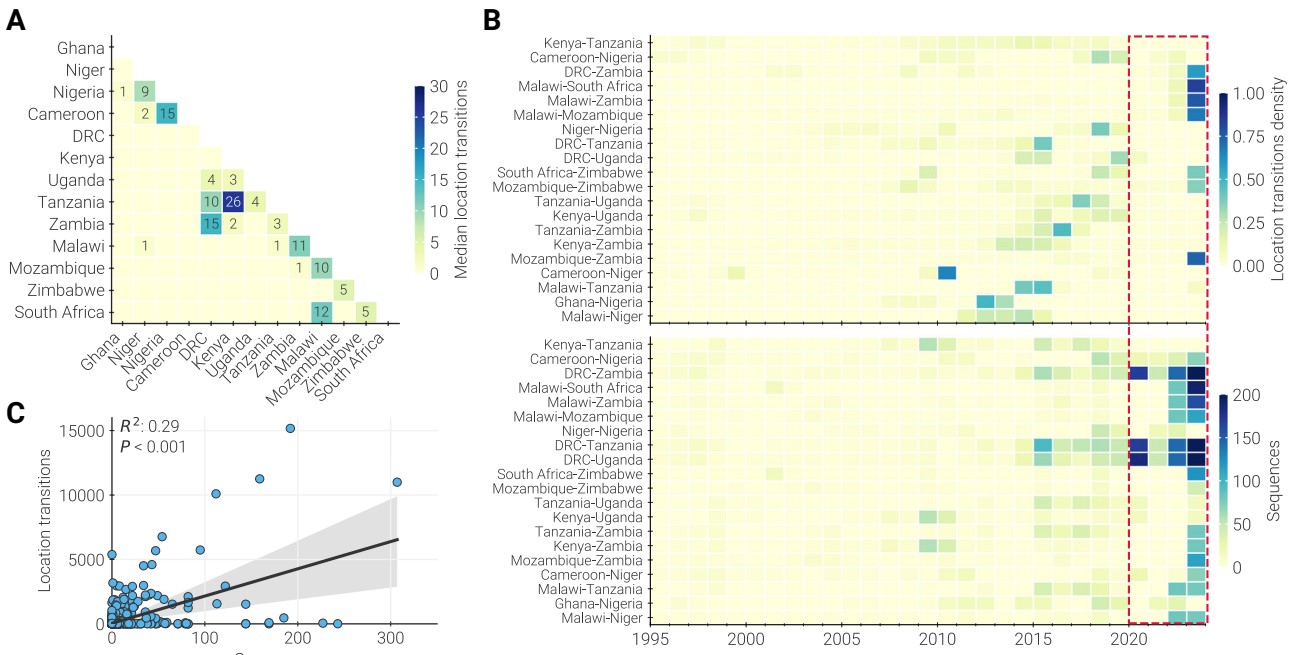

**Fig. 3 | Dynamics of cross-border transmission. A** Median number of location transitions observed between each pair of countries across the posterior distribution of trees from the discrete state analysis. Countries were included as states in the discrete state reconstruction if they had more than 20 cholera genomes in our dataset (see Methods). Only location pairs with a supported transition rate in at least 50% of the posterior are labeled. Countries are ordered west-to-east, north-to-south, and so that neighboring countries are next to each other. **B** Top panel: annual density of location transitions for each location pair with a supported transition rate in at least 50% of the posterior. Order of locations in each pair is alphabetical and does not reflect directionality. Bottom panel: annual number of genomes available for each location pair. Red box highlights period of increased sampling (due to CholGEN) corresponding to higher density of observed location transitions. **C** Correlation between the number of genomes available and the magnitude of location transitions for a given year and location pair. $R^2$ and $P$ value were calculated using ordinary least squares regression. The P value represents the results of a two-tailed student T test where the null hypothesis is a slope of zero. Shaded error band for linear regression represents 95% confidence interval determined using 1000 bootstrap resamples.

These findings point to possible intercontinental spread of cholera strains carrying an IncA/C plasmid and support the connection between outbreaks in the Middle East and Africa.

### Cross-border transmission is frequent but rarely observed

The results above suggest that transmission patterns, rather than genotypic differences, may be crucial for understanding and preventing future cholera spread. The recency of collected isolates and close proximity of the CholGEN AU Member States, including those with ongoing AFR15 outbreaks, enabled us to take a close look at the location and frequency of transmission of AFR15 and other lineages across international borders. Highlighting the importance of regional-coordination, we identified numerous examples of cross-border transmission across all lineages currently circulating in Africa.

To closely examine cross-border transmission, we quantified the timing and number of geographic transitions (also called Markov jumps[45]), between AU Member States across the full posterior of a Bayesian phylogeographic reconstruction. While we found multiple prominent examples of cross border transmission in the phylogeny (Fig. 3A), our initial analysis suggested that these transitions were infrequent during the third wave of 7PET (average rate of 0.04 location transitions per year; 95% HPD: 0.03–0.05 transitions per year). However, historical sampling of isolates has been inconsistent and likely biased, and we found that location transitions were unevenly distributed over time (Fig. 3B). Closer inspection revealed that the timing of location transitions was moderately correlated with sampling ($R^2$ value = 0.29; p-value < 0.001; Fig. 3C). This result suggests that international transmissions may be more frequent and consistent than what was captured in the dataset, but only observed whenever genomic surveillance was adequate. Furthermore, this means that transmission

involving unsampled or undersampled locations are underappreciated in phylogenetic analyses. Therefore, we expect that increased and expanded surveillance may identify locations with yet-unsampled outbreaks, allowing for more effective and accurate targeting of containment efforts to locations that are continuously reseeding transmission.

### Future surveillance should be directed towards high diversity locations

Our results indicate that sampling plays an important role in observing cholera transmission and suggest that moving towards routine genomic surveillance may support international cholera elimination goals by revealing transmission patterns that are currently missed. Therefore, we attempted to estimate un-captured transmission and determine if additional surveillance is equally needed everywhere or if there are certain lineages, countries, or regions that are most in need of additional sampling. To do so, we developed a framework to assess the value of generating a new genome that involves quantifying the sampling strategy, phylogenetic diversity, and total information captured by sequencing.

We took this holistic approach to assessing the value of generating new sequences because both sampling strategy and underlying transmission patterns influence what can be learned from sequencing. To evaluate the impact of the sampling strategy, we built on a technique commonly used in ecology[46,47] and estimated how many new mutations would be expected when adding a new genome to the analysis, referred to here as phylogenetic diversity[48,49]. While this value provides an estimate of the potential information gained from additional sequencing, it must also be considered in the context of the total number of sequences previously generated. We also recognize that

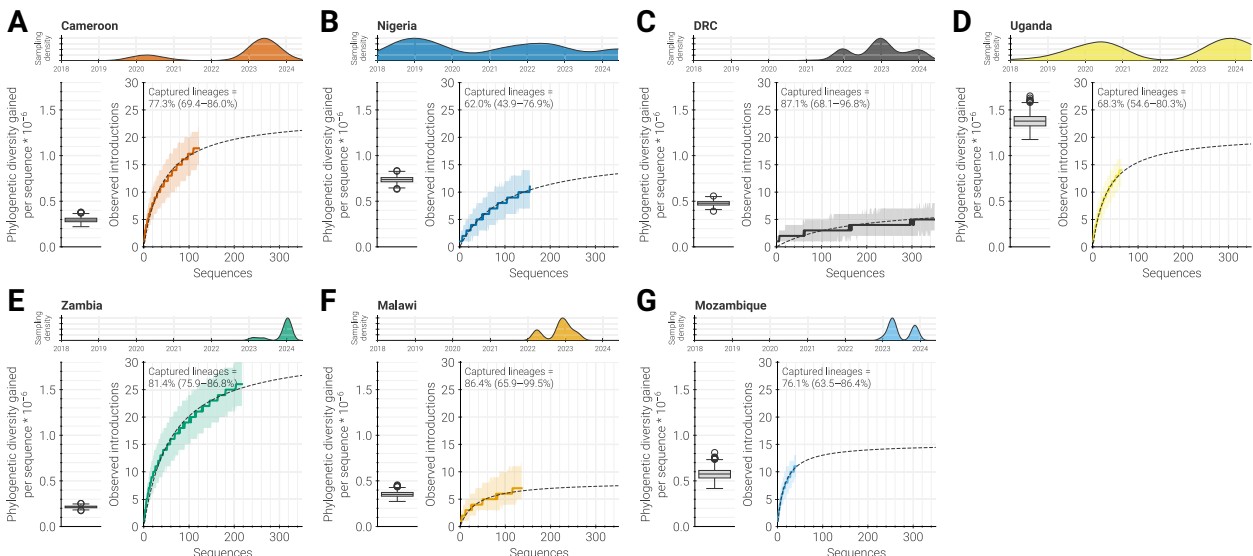

**Fig. 4 | Assessing current surveillance.** For each CholGEN Member State **A–G**, metrics that estimate the value of sequencing new *V. cholerae* isolates. Top: temporal distribution of genomic surveillance conducted by CholGEN within each Member State. Density curves for each country are normalized so that they have the same maximum value (see Fig. 1B for actual values). Lower left: substitutions per site gained by each genome sequenced by CholGEN (distribution across each tree in the posterior distribution of the phylogeographic analysis). For each country, the box extends from the first quartile to the third quartile of the phylogenetic diversity observed, with a line at the median. Whiskers extend to the farthest data point within 1.5 inter-quartile ranges from the box. Lower right: rarefaction curves describing the rate at which introductions into the country are observed given the number of genomes collected. A solid line and shaded area represents the median rarefaction curve and its 95% confidence interval, while the dashed black line represents the Michaelis-Menten model fit. Distributions for rarefaction curves and percentages of total introductions that were observed represent 1000 random trials wherein a tree was independently sampled from the posterior and a rarefaction curve calculated. The percentage of total introductions that were observed and 95% confidence intervals are recorded in the upper right corner of each country's panel.

sequencing can still be useful in cases of limited phylogenetic diversity if there is other information to be gained, for example if additional sequencing may reveal new introductions into a country or region. Therefore, we used a rarefaction approach[50,51] to gauge how much we would likely learn about introductions from additional sequencing.

It is important to note that these estimates are relevant only to the period during which surveillance was conducted. These values will inevitably change if epidemiological, climate, and social conditions shift in the future. Therefore, this framework uses these metrics as guides to direct surveillance efforts rather than rigid sequencing requirements, which must be considered alongside other relevant information.

We used this framework to assess the genomic data generated by each CholGEN Member State and saw that the amount of diversity and information gained per new genome varied between locations and surveillance intensities (Fig. 4A–G). Below, we highlight three examples—Uganda, DRC, and Zambia—that illustrate the range of phylogenetic diversity captured and additional information revealed given the differing surveillance strategies employed.

First, new genomes from Uganda revealed marked phylogenetic diversity and numerous introductions (Fig. 4D). This result is expected, as CholGEN samples from Uganda cover multiple years and longer sampling periods should capture more diversity assuming a constant clock rate[52]. It also aligns with transmission dynamics previously described in Uganda, where repeated introductions (rather than local spread) sustain cholera and thereby increase phylogenetic diversity[24,53]. These findings suggest that continued surveillance using a similar strategy, where genomic data is consistently collected over time, would likely reveal further introductions and greater diversity. This could help identify additional sources of cholera or new transmission routes that could be targeted for intervention.

In contrast to Uganda, new genomes from DRC revealed relatively less phylogenetic diversity and fewer introductions, though we

estimated that there are far fewer introductions into DRC than into Uganda (Fig. 4C). This is likely due to distinct transmission patterns in DRC, where cholera is highly persistent (Fig. 2B). These results suggest that further surveillance may uncover more phylogenetic diversity, but it is unlikely to reveal new introductions without a change in the surveillance strategy or overall transmission pattern within the country. Therefore, routine surveillance will most likely provide a representative view of the situation in DRC without requiring intensive sampling of every outbreak. However, it will be important to ensure that sampling in DRC is representative across space in addition to over time. For instance, if surveillance is not geographically representative, it may be useful to first capture geographic gaps to ensure lineages are not being missed before finalizing a surveillance strategy.

In contrast to Uganda and DRC, new genomes from Zambia showed comparatively low levels of phylogenetic diversity. However, Zambia is an example of a situation in which phylogenetic diversity was low but the potential to capture additional information (for instance, number of introductions) was very high, as new data uncovered many additional introductions despite genetic similarity (Fig. 4E). This finding aligns with the relatively limited range of collection dates for Zambia sequences generated by CholGEN, as well as with evidence for the close transmission relationships of Zambia with Malawi, DRC, and other locations. To better capture the diversity of circulating lineages and the locations that seed cholera in Zambia, additional surveillance would need to be conducted over a longer period. Additionally, during periods of rapid spread by a single lineage, resources may be better spent capturing temporal or geographic diversity rather than sequencing large numbers of likely identical sequences from concentrated areas or clonal outbreaks.

## Discussion
In this manuscript, we highlight how the CholGEN initiative led to the enhancement of local cholera genomic surveillance in several AU

Member States, leading to the largest dataset of high-quality *V. cholerae* genomes sequenced locally in Africa. These genomes fill critical gaps in our understanding of recent cholera circulation, revealing distinct transmission patterns across different African regions and highlighting the rapid spread of the AFR15 lineage, including its expansion into additional Member States like the DRC. Despite the association of AFR15 with unusually large outbreaks in Malawi, Zambia, and beyond, we did not identify significant genotypic differences compared to other lineages previously identified in the region. Antimicrobial resistance signatures were also consistent across lineages and outbreaks. We found evidence of cross-border transmission between neighboring countries, and our data suggests that these events may be more frequent than currently recognized, which could have significant implications for how local experts develop surveillance strategies and cholera control efforts.

The proximity of CholGEN Member States and their distribution across Africa allowed us to compare the spread of cholera within and between different regions, particularly for the most recent outbreaks on the continent. This work provides further evidence that cholera readily spreads within certain regions of Africa, indicating that cross-border transmission has a prominent role in the maintenance of cholera on the continent. That said, these analyses clearly show that sampling strategy plays a significant role in the interpretation of phylogenetic analyses. In particular, transmission involving unsampled or undersampled locations may be underappreciated, and our ability to comment on the directionality or absolute number of international transmissions, both within Africa and from other regions into Africa, is limited. Greater availability of global sequence data would help clarify major transmission events, such as the AFR15 introduction, and regional transmission patterns. Refining our understanding of cholera transmission will require continuous and routine genomic surveillance, both during and between cholera outbreaks, and in all countries with reported cholera infections. This will both fill in gaps in our current understanding and provide a more systematic dataset that can inform future prioritization of surveillance through the sequencing information metrics described in Fig. 4 above. By evaluating these metrics in the context of the transmission patterns present in a country, surveillance efforts can be more effectively tailored to meet specific needs and optimally allocate resources.

It is important to recognize, however, that the real barriers to sampling are often not in the strategy, but in the realities of sample collection, transport, sequencing, and pairing with epidemiological metadata. Although all sequencing described in this manuscript was performed from cultured isolates, we were only 60% effective in generating high-quality whole genome sequences. This is likely due to a combination of isolate contamination, storage conditions not being conducive to preserving bacterial cultures, and other challenges associated with multi-step processes like whole genome sequencing. To enable better surveillance of cholera and other local pathogens, we need to improve the full pathway leading to sequencing, from sample collection and *V. cholerae* identification through sample transport and laboratory processing. Additionally, it may be important to explore sequencing methods that do not rely on bacterial culture, such as direct sequencing from stool[17,54]. This will increase the total number of samples that can be selected for sequencing and open the door to improved environmental surveillance including viable but nonculturable specimens that may play an important role in understanding transmission[55]. While these direct sequencing approaches would likely require increased sequencing depth (and therefore, per-sample cost) and analysis complexity; the additional information gained would be valuable, necessitating careful cost-benefit evaluation of such methods.

Even once sequencing challenges are addressed, there are important next steps to take in terms of translating findings to public health impact. For example, we found that isolates in Uganda acquired a plasmid associated with AMR, but without phenotypic data paired to these samples, it is difficult to understand the implications of such an acquisition. Further in-depth studies comparing antimicrobial resistance phenotype to genotype[38] are needed to assess which genotypes correlate with actual antibiotic resistance, and which phenotypic changes are caused by de novo evolution of resistance-conferring mutations versus the acquisition of mobile genetic elements from other bacterial organisms (*V. cholerae* and otherwise) through horizontal gene transfer. This information would provide a clear and actionable benefit to public health, as genotypic surveillance could therefore also serve the role of current antimicrobial susceptibility assays, enabling the use of one holistic assay (whole genome sequencing) to obtain the results of what is currently at least two separate experiments.

Additionally, while our work provides further evidence of cross-border transmission of cholera, coordination with local epidemiology and policy teams is needed to figure out how this transmission is happening and how best to stop it. Regional analyses, as demonstrated here, can identify broad transmission patterns, such as our finding that transmissions mostly occur between neighboring countries. However, more local, targeted investigations using sequencing to confirm epidemiological hypotheses are needed to determine fine-scale patterns that can more directly enhance containment and intervention strategies. Future studies could also address the effectiveness of interventions directly by examining how WASH implementation affects circulating cholera diversity or by investigating mutations in the O-antigen gene cluster and cholera toxin genes during oral vaccine campaigns (as immune responses against the O-specific polysaccharide play a critical role in protective immunity to cholera[56,57] and recombinant cholera toxin B subunit is a component in some oral cholera vaccines).

Through the creation of CholGEN, we have created a collaborative, multicountry group working to advance cholera containment and elimination through genomics-informed decision making. In this manuscript, we showcase how developing and extending local sequencing and analytical capacity enabled us to elucidate recent patterns in *V. cholerae* spread and evolution. Our next steps include more in-depth analyses of the collected data, including exploration of virulence factors, vaccine targets, and within-country cholera spread. The genomic dataset we generated will inform prospective surveillance in all seven AU Member States using the sampling assessment framework we proposed to ensure representative sequencing of isolates. To support these initiatives, we will continue to provide training programs and computational infrastructure to strengthen bioinformatics capacity in Africa. Coupled with effective data sharing and collaboration, we hope that these efforts, and the conclusions we draw from them, will bring us one step closer to the goal of global cholera elimination by 2030.

## Methods

### Sample collection, cholera confirmation, and bacterial culture

*V. cholerae* isolates were collected from clinically suspected cholera cases in CholGEN Member States between 2018 and 2024. In each Member State, isolates came from patients from cholera treatment centers during outbreaks or from endemic areas. In all seven countries, the Cholera Rapid Diagnostic Test positive samples were transported in Cary-Blair media to laboratories then cultured on Thiosulfate-citrate-bile salts-sucrose (**TCBS**) agar medium and incubated for 20–24 hours. Phenotypic identification of *V. cholerae* colonies was based on morphology, motility, and biochemical characteristics (positive oxidase, saccharose, indole, and gelatinase). *V. cholerae* isolates were confirmed with agglutination tests with anti-O1 or anti-O139 serum (WHO antisera).

## Genomic DNA extraction and quantification

DNA extraction was performed at multiple laboratories across the CholGEN participating countries: NPHL (Yaounde, Cameroon), CPHL (Kampala, Uganda), INS (Maputo, Mozambique), ZNPHI (Lusaka, Zambia), PHIM (Lilongwe, Malawi), INRB (Kinshasa, DRC), and NCDC (Abuja, Nigeria). Confirmed *V. cholerae* isolates were retrieved from storage and subcultured on selective TCBS, MH, and/or HCK agar plates and incubated at 37 °C overnight. In all countries, *V. cholerae* DNA was extracted using the standard protocol from the Qiagen QIAamp DNA Mini Kit, with a final elution volume of 200 μL as per the manufacturer's instructions. Genomic DNA concentrations were measured with the Qubit Fluorometer 4.0 (Thermo Fisher), following the dsDNA HS assay standard protocol and stored at 4 °C.

## Library preparation and sequencing

Illumina library preparation and sequencing was performed by laboratories in CholGEN participating countries. DNA samples were normalized to 0.6 ng/μL from which 30 μL was used as input for the Illumina DNA Prep Kit (Illumina). Library concentrations were assessed using the Qubit High Sensitivity DNA Kit (Invitrogen), and library size distributions were measured using a BioAnalyzer High Sensitivity DNA Kit (Agilent). Genomic libraries from individual samples that had expected size distributions and had a concentration greater than 1 ng/μL were normalized and pooled in equimolar amounts at 2 nM. The 2 nM library pool was sequenced on either an Illumina MiSeq or NextSeq 2000 using 300 cycles kits. Quality assessments of short reads were performed using FastQC[58].

## Reference-based Genome Assembly

We developed a pathogen-agnostic bioinformatics pipeline to generate consensus sequences from raw sequencing reads, called *bacpage*, which is available as a local command line tool (available on Github: https://github.com/CholGen/bacpage; with an online manual: https://cholgen.github.io) on the cloud-based Terra platform[59], where a majority of analyses were performed. Briefly, as part of the pipeline, paired-end reads were aligned against the *V. cholerae* N16961 isolate (accession AE003852 /AE003853 [https://www.ncbi.nlm.nih.gov/nuccore/AE003853]) using bwa-mem v0.7.17[60]. Variants compared to the reference were identified using BCFTools v1.20[61]. Variants were retained if they met the following criteria: (1) variant quality score of at least 20, (2) mapping score of at least 30, (3) supported by at least 15 reads, (4) present in at least 50% of reads covering a position, and (5) supported from both strands. A consensus sequence for each sample was generated by applying the supported variants to a concatenated reference genome.

Consensus sequences for each sample were kept for subsequent analyses if at least 90% of their reads mapped to the reference genome, had median coverage across all positions of the reference genome of at least 15, and had less than 10% ambiguous nucleotides.

## Vibrio cholerae Genomic Data

The sequences generated in this study were combined with 1772 publicly available third wave (AFR9 onwards) genomes that were found by searching literature, Sequence Read Archives (**SRA**), and VibrioWatch for *V. cholerae* isolates from Africa and Asia (Supplementary Data 2). Where raw sequencing data was available, we generated reference-based assemblies using the same pipeline and reference as described above. In other cases, where only assembled contigs were available, variants compared to the N16961 reference were identified using snippy v4.6.0 (https://github.com/tseemann/snippy) with the --ctgs flag and applied to the reference genome using BCFTools. Lastly, we filtered identical sequences from the dataset to reduce the computational burden of downstream analyses. When identical sequences were found from the same country and collection month, only one sequence was randomly selected to be kept in the dataset.

## Antimicrobial Resistance Profiling

Short reads were assembled de novo using the TheiaProk Illumina paired-end sequencing workflow publicly available on DockStore[62]. Briefly, the workflow trims low quality reads and portions of reads using fastp[63] and removes sequencing adaptor sequences using BBDuk[64]. Assembly was performed on high-quality reads with the Shovill pipeline (https://github.com/tseemann/shovill) which uses the SKESA assembler[65]. We then used AMRFinderPlus v3.11.20 with database version 2023-09-26.1 to identify antimicrobial resistance genes and point mutations in the assemblies[66].

## Epidemiological Data

We obtained yearly case counts for each country participating in CholGEN from data curated by the cholera taxonomy project at the Johns Hopkins Bloomberg School of Public Health (https://cholera-taxonomy.middle-distance.com/). Data contributions to the team have come from many sources, though the annual case counts are heavily dependent on the WHO Annual Cholera Reports and the Global Health Observatory data repository.

## Phylogenetic Analysis

We used the `phylogeny` subcommand of bacpage to generate a recombination-masked maximum likelihood phylogeny. Briefly, the pipeline performs the following steps. The complete genomic dataset was assembled by concatenating the individual reference-based assemblies along with the N16961 reference into a pseudo-alignment. Problematic sites including known recombinant regions, repetitive sequences, and homoplastic sites were masked based on the GFF file provided by Weill et al. [2]. Novel recombinant regions and regions with significantly elevated densities of substitutions were masked using gubbins v2.3.4[67]. To reduce the computational burden of the phylogenetic analysis, we filtered the pseudo-alignment to only include variable positions using SNP-sites v2.5.1[68]. We constructed a maximum likelihood phylogenetic tree for the dataset using IQ-TREE2 and an GTR substitution model[69,70].

The resulting phylogeny was rooted on the reference genome. To further reduce computation burden of the analysis we extracted the subtree descending from the most recent common ancestor of lineages AFR9-15, which corresponds to the third wave of 7PET. The subtree was time-resolved using Treetime v0.11.3[71]. Taxa deviating more than three interquartile distances from the clock-rate regression were pruned from the phylogeny. Additionally, we randomly resolved polytomies in the phylogeny by adding zero-length branches with gotree[72].

We refined the time-resolved phylogeny using BEAST v1.10.5[73]. We specified a GTR substitution model with gamma distributed rate heterogeneity under an uncorrelated relaxed molecular clock and a constant coalescent tree prior. For the uncorrelated relaxed molecular clock, we specified an informative lognormal distribution with a mean of $8.4*10^{-7}$ substitutions/site/year and a standard deviation of $1.4*10^{-6}$ consistent with Weill et al.[2]. To account for utilizing an SNP alignment rather than a full-genome alignment, we additionally supplied the number of constant sites in the cholera reference genome.

We ran two MCMC chain of 300 million steps utilizing the BEAGLE computational library[74]. Parameters and trees were sampled every 10,000 and 100,000 steps, respectively. Convergence and mixing of the MCMC chains were assessed with Tracer v1.7.2 and Beastiary v1.8.3[75,76]. We confirmed that all estimated continuous parameters had effective sample sizes greater than 200 and all key summary statistics had effective sample sizes greater than 150. BEAST XML and log files for the phylogenetic analysis are available on GitHub (https://github.com/CholGen/RegionalAnalysis-2024).

## Lineage Assignment

African *V. cholerae* isolates are generally classified by the introduction event they descend from (of which there have been ~17 described).

Weill et al. initially delimited these introductions by performing a discrete state reconstruction of the sampling location of tree tips; with introductions being defined as discrete state transition from "Asia" to "Africa." Because the clades descending from these introductions are necessarily monophyletic, they are easily identifiable by visual inspection of the tree. This means the nomenclature for African isolates is based on phylogenetic placement, rather than a genotype (at least not explicitly). Assigning sequences to a lineage therefore encompasses including the new sequences in a phylogeny, after which they can be assigned a "lineage" based on the lineage clade they are placed in.

Due to the large number of sequences generated by CholGEN, manual assignment was not feasible. Instead, for each lineage, we identified the node corresponding to the most recent common ancestor (**MRCA**) of lineage representatives collected from prior publications[2,16,26,28,77]. All sequences descending from the MRCA, i.e. fall within the lineage's clade, without a lineage assignment were assigned to the lineage. Sequences that were collected from Africa but did not descend from a previously described lineage were assigned a value of "sporadic."

To avoid the high computational cost of a complete phylogenetic reconstruction for lineage assignment, we developed a tool to identify the lineage corresponding to a newly sequenced *V. cholerae* genome called Vibecheck. Briefly, the tool uses Ultrafast Sample Placement on Existing tRees (**UShER**) to perform rapid parsimony-based phylogenetic placement of new sequences in a mutation- and lineage-annotated global *V. cholerae* phylogeny[78]. Vibecheck is available along with its documentation on Github (https://github.com/CholGen/Vibecheck).

### Mutation Profiling

We performed ancestral state reconstruction on the O1 *V. cholerae* ML phylogeny using the ancestral subcommand of the augur bioinformatics toolkit[79]. W parsed out reconstructed sites and cataloged the type single nucleotide mutations that occurred on the branches of each lineages' subtree using a custom script. Comparisons between the proportion of each mutation type (ex. A- > T, G- > C, etc.) for each third-wave lineage versus the proportion observed on non-lineage branches was conducted using a Pearson's chi-squared test. Non-synonymous mutations were determined and translated using the gene coordinates from the N16961 *V. cholerae* isolate (accession AE003852/AE003853). Mutated genes were mapped to gene ontology biological processes using the PANTHER API[80–82]. We additionally used the PANTHER API to determine whether any gene sets mutated across a lineage's branches were statistically overrepresented[83].

### Phylogeographic Reconstruction

We performed a discrete state ancestral reconstruction on geographic states using BEAST[45]. This analysis reconstructed location-transition history across an empirical distribution of 1000 time-calibrated trees samples from the posterior tree distribution estimated above[84,85]. To conduct the analysis, we assigned each taxon a discrete location state based on the country where it was collected. To limit the number of transition rates needing to be estimated, we binned non-African sequences into a single "Other" state, and African countries with less than 20 reported sequences into a single "Other-Africa" state. Ultimately we used 14 distinct locations states. We assumed that geographic transition rates were reversible and used a symmetric substitution model. We used Bayesian stochastic search variable selection to infer non-zero migration rates[45]. The MCMC algorithm was run for 500,000 generations and sampled every 500 generations. We used the TreeMarkovJumpHistoryAnalyzer from the pre-release version of BEAST v1.10.5 to obtain the Markov jump estimates and their timings from the posterior tree distribution and assumed that they are a suitable proxy for the transmission between two locations. We used

TreeAnnotator v1.10 to construct a maximum clade credibility (**MCC**) tree which we visualized with baltic (https://github.com/evogytis/baltic). BEAST XML and log files for the discrete phylogeographic analysis are available on GitHub (https://github.com/CholGen/RegionalAnalysis-2024).

### Introduction Identification

We identified country-specific strains from the posterior tree distribution of the discrete phylogeographic reconstruction using a custom script. Country-specific strains were defined as two or more taxa from the same country that descend from a single, shared introduction of cholera into the country from another country. Descendant taxa from outside the specific country, do not belong to the strain. This concept and the algorithm for identifying strains is derived from the definition of Du Plessis et al.[86] (see the citation for an illustration of this concept, wherein the term transmission lineage is used in place of strain).

### Phylogenetic diversity

We calculated the amount of evolutionary history contributed by a set of genomes as their phylogenetic diversity[49,87,88]. For each tree in the posterior distribution of the phylogenetic analysis, we converted the units of branch lengths from years to substitutions/site by multiplying branch lengths by the estimated substitution rate for each branch. For each country, we marked taxa that were collected from the country and were generated by CholGEN participants. We then calculated the phylogenetic diversity contributed by CholGEN for a given country as the difference between the total branch length of the tree and the total branch length of the tree after pruning the marked taxa. Lastly, this value was divided by the total number of sequences marked for a given country.

### Rarefaction analysis

For each country, we identified unique introductions into each country and identified genomes that descended from those introductions for each tree in the posterior. For varying sample sizes, ranging from 1 to the total number of genomes collected for a country, genomes were randomly drawn without replacement and the number of unique introductions in the sample was computed. To incorporate uncertainty from the discrete state reconstruction, we performed this rarefaction analysis 1000 times, using a randomly sampled tree each time. For each trial, we parametrically estimated the total number of introductions into each country by fitting the rarefaction curve with a Michaelis-Menten equation. The Michaelis-Menten equation is a two parameter model with the following form:

$$y = \frac{V^*x}{K+x} \qquad (1)$$

where $x$ is the number of genomes sampled, $y$ is the number of introductions observed, $V$ is the asymptotic estimate of the total number of introductions, and $K$ is a shape parameter describing the rate at which new genomes reveal introductions.

### Ethics statement

This research utilized samples collected through routine national surveillance programs and all data and samples were anonymized to ensure the privacy and confidentiality of the individuals involved. The use of these samples was approved by respective Ethical Review Boards/Institutions in each country, ensuring that the research adhered to the highest ethical standards and legal requirements. In Cameroon, ethical approval was granted by the National Committee on Ethics in Research for Human Health (2024/02/1640/CE/CNERSH/SP). In DRC, ethical approval was granted by the Board of the Ethics Committee of the School of Public Health at the University of Kinshasa

(ESP/CE/148/2023 and ESP/CE/149/2023). In Malawi, this work was approved by the National Health Sciences Research Committee (Protocol #867). In Mozambique, ethical approval was granted by the National Bioethics Committee for Health (335/CNBS/23). In Nigeria, ethical approval was not required as it is based on data from Nigeria's national surveillance program, collected by the Nigeria Centre for Disease Control. In Uganda, ethical approval was granted by the Uganda Ministry of Health National Health Laboratory Services (UNHL-2024-88). In Zambia, ethical approval was granted by the University of Zambia Biomedical Research Ethics Committee (UNZA-7540/2025).

### Reporting summary
Further information on research design is available in the Nature Portfolio Reporting Summary linked to this article.

## Data availability
Raw sequencing reads are available on NCBI under the BioProject accession ID PRJNA1145341 and Sequence Read Archive accession IDs are provided in Supplementary Data 1. Accession IDs for the publicly available sequences acquired from NCBI or VibrioWatch are provided in Supplementary Data 2.

## Code availability
Code for all analyses and figure generation are available at: https://github.com/CholGen/RegionalAnalysis-2024[89]. Links to XMLs and log files for BEAST analyses can be found within the Github repository.

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

## Acknowledgements

We extend our sincere gratitude to the Africa Centres for Disease Control and Prevention (Africa CDC), Africa Public Health Foundation (APHF), and Broad Institute of MIT and Harvard, for their invaluable support and collaboration. This work was supported by the Gates Foundation grants INV-047157 (S.K.T.)] and INV-047156 (A.K.D. and D.A.S.).

## Author contributions

G.M., N.L.M., C.K.T., S.W., and S.K.T. conceptualized the study. G.M., N.L.M., A.S.A., and S.W. contributed the methodology. N.L.M. provided software. G.M. and N.L.M. conducted the formal analysis. M.K., G.K.K., O.A.A., J.J.C., Mathews Kagoli, R.G.E., A.A., I.S., V.J.N., A.A.A., O.A.B., B.M.A., A.B., E.B.M., F.O., C.E.C., N.I., O.K., O.L., P.K.M., G.A.E., I.C.M., A.A.M., G.N., J.M., K.M., A.M.N., M.E.N., M.P., D.M.S., C.M., S.M.S., D.W.W., P.O.W., M.Y., S.Y., L.Z., and CholGEN Consortium authors provided resources. G.M., M.K., G.K.K., O.A.A., J.J.C., Mathews Kagoli, R.G.E., A.A., I.S., V.J.N., A.A.A., O.A.B., B.M.A., A.B., E.B.M., F.O., C.E.C., N.I., O.K., O.L., P.K.M., G.A.E., I.C.M., A.A.M., G.N., J.M., K.M., A.M.N., M.E.N., M.P., D.M.S., C.M., S.M.S., D.W.W., P.O.W., M.Y., S.Y., L.Z., and CholGEN Consortium authors curated data. G.M., N.L.M., C.K.T., S.W., and S.K.T. wrote the original draft of the manuscript. All authors reviewed and edited the manuscript. N.L.M. performed visualization. J.E.B., R.C., H.H., J.I., J.P.L., D.M., S.N., A.K.D., D.A.S., J.K., Y.K.T., S.W., and S.K.T. supervised the study. G.M., C.K.T., S.W., and S.K.T. undertook project administration. A.K.D., D.A.S., and S.K.T. acquired funding.

## Competing interests

The authors declare no competing interests.

## Additional information

Gerald Mboowa [1,27], Nathaniel Lucero Matteson [2,27], Collins Kipngetich Tanui [1,27], Mpanga Kasonde [3], Guyguy Kusanzangana Kamwiziku [4,5], Olusola Anuoluwapo Akanbi [6], Jucunú Johane Elias Chitio [7], Mathews Kagoli [8], René Ghislain Essomba [9,10], Alisen Ayitewala [11], Isaac Ssewanyana [11], Valentina Josiane Ngo Bitoungui [9,12], Adrienne Aziza Amuri [5,13], Andrew S. Azman [14,15,16], Olajumoke Atinuke Babatunde [6], Blaise Mboringong Akenji [9], Anaïs Broban [17], Espoir Bwenge Malembaka [14,18], Francis Ongole [11], Chimaobi Emmanuel Chukwu [6], Nália Ismael [7], Otridah Kapona [3], Osvaldo Laurindo [7], Placide Kingebeni Mbala [4,5], Georges Alain Etoundi Mballa [19], Imelda Carlos Zulfa Miambo [7], Alex Ansaye Mwanyongo [8], Grace Najjuka [11], Joseph Mutale [3], Kunda Musonda [3], Allan Muruta Niyonzima [20], Mirriam Ethel Nyenje [8], Michael Popoola [6], Doreen Mainza Shempela [21], Christiane Medi Sike [10,22], Sofião Manjor Sitoe [7], Dorcas Waruguru Wanjohi [1], Placide Okitayemba Welo [23], Mtisunge Yelewa [8], Sebastian Yennan [6], Lucius Ziba [8], CholGEN Consortium*, Joseph Ephram Bitilinyu-Bangoh [8], Roma Chilengi [3], Hamsatou Hadja [9,24], Jide Idris [6], José Paulo Maurício Langa [7], Daniel Mukadi-Bamuleka [25], Susan Nabadda [11], Amanda K. Debes [26], David A. Sack [26], Yenew Kebede Tebeje [1], Shirlee Wohl [2,28] ✉ & Sofonias Kifle Tessema [1,28] ✉

[1]Africa Centres for Disease Control and Prevention, Addis Ababa, Ethiopia. [2]Division of Infectious Diseases, Brigham and Women's Hospital, Boston, USA. [3]Zambia National Public Health Institute, Lusaka, Zambia. [4]Department of Medical Biology, University of Kinshasa, Kinshasa, Democratic Republic of the Congo. [5]Institut National de Recherche Biomédicale, Kinshasa, Democratic Republic of the Congo. [6]Nigeria Centre for Disease Control and Prevention, Abuja, Nigeria. [7]Instituto Nacional de Saúde, Marracuene, Mozambique. [8]Public Health Institute of Malawi, Lilongwe, Malawi. [9]National Public Health Laboratory, Ministry of Public Health, Yaoundé, Cameroon. [10]Faculty of Medicine and Biomedical Sciences, University of Yaounde 1, Yaoundé, Cameroon. [11]National Health Laboratory & Diagnostic Services (NHLDS/CPHL), Ministry of Health, Kampala, Uganda. [12]Faculty of Medicine and Pharmaceutical Sciences, University of Dschang, Dschang, Cameroon. [13]Faculty of Medicine, University of Kinshasa, Kinshasa, Democratic Republic of the Congo. [14]Department of Epidemiology, Johns Hopkins Bloomberg School of Public Health, Baltimore, USA. [15]Geneva Centre for Emerging Viral Diseases, Geneva University Hospitals,

Geneva, Switzerland. [16]Division of Tropical and Humanitarian Medicine, Geneva University Hospitals, Geneva, Switzerland. [17]Epicentre, Paris, France. [18]Center for Tropical Diseases and Global Health, Université Catholique de Bukavu, Bukavu, Democratic Republic of the Congo. [19]Department for the Control of Diseases, Epidemics and Pandemics, Ministry of Public Health, Yaoundé, Cameroon. [20]Integrated Epidemiology, Surveillance and Public Health Emergencies, Ministry of Health, Kampala, Uganda. [21]Churches Health Association of Zambia, Lusaka, Zambia. [22]Hôpital Laquintinie de Douala, Douala, Cameroon. [23]National Program of Elimination of Cholera, Ministry of Health, Kinshasa, Democratic Republic of the Congo. [24]Yaoundé General Hospital, Yaoundé, Cameroon. [25]Rodolphe Merieux Laboratory INRB-Goma, Goma, Democratic Republic of the Congo. [26]Department of International Health, Johns Hopkins Bloomberg School of Public Health, Baltimore, USA. [27]These authors contributed equally: Gerald Mboowa, Nathaniel Lucero Matteson, Collins Kipngetich Tanui. [28]These authors jointly supervised this work: Shirlee Wohl, Sofonias Kifle Tessema. *A list of authors and their affiliations appears at the end of the paper. ✉e-mail: swohl@bwh.harvard.edu; SofoniasT@africacdc.org

## CholGEN Consortium

Gerald Mboowa [1,27], Nathaniel Lucero Matteson[2,27], Collins Kipngetich Tanui[1,27], Mpanga Kasonde[3], Guyguy Kusanzangana Kamwiziku [4,5], Olusola Anuoluwapo Akanbi [6], Jucunú Johane Elias Chitio[7], Mathews Kagoli[8], René Ghislain Essomba[9,10], Alisen Ayitewala[11], Isaac Ssewanyana[11], Valentina Josiane Ngo Bitoungui[9,12], Adrienne Aziza Amuri [5,13], Olajumoke Atinuke Babatunde[6], Blaise Mboringong Akenji[9], Francis Ongole[11], Chimaobi Emmanuel Chukwu [6], Nália Ismael[7], Otridah Kapona[3], Osvaldo Laurindo[7], Placide Kingebeni Mbala [4,5], Georges Alain Etoundi Mballa[19], Imelda Carlos Zulfa Miambo[7], Alex Ansaye Mwanyongo[8], Grace Najjuka[11], Joseph Mutale[3], Kunda Musonda[3], Allan Muruta Niyonzima[20], Mirriam Ethel Nyenje[8], Michael Popoola[6], Doreen Mainza Shempela[21], Christiane Medi Sike[10,22], Sofião Manjor Sitoe[7], Dorcas Waruguru Wanjohi[1], Placide Okitayemba Welo[23], Mtisunge Yelewa[8], Sebastian Yennan[6], Lucius Ziba[8], Joseph Ephram Bitilinyu-Bangoh[8], Roma Chilengi [3], Hamsatou Hadja[9,24], Jide Idris[6], José Paulo Maurício Langa[7], Daniel Mukadi-Bamuleka [25], Susan Nabadda[11], Amanda K. Debes[26], David A. Sack [26], Yenew Kebede Tebeje[1], Shirlee Wohl [2,28] ✉ & Sofonias Kifle Tessema [1,28] ✉

A full list of members and their affiliations appears in the Supplementary Information.

