## [Transparent Peer Review file · Nature Communications]

Multicountry genomic analysis underscores regional cholera spread in Africa

Corresponding Author: Dr Sofonias Kifle Tessema

Version 0:

Reviewer comments:

Reviewer #2

(Remarks to the Author)

The authors have satisfactorily addressed my previous concerns. They have expanded the dataset, added analyses, acknowledged sampling biases, and clarified language. While some conclusions remain high-level, the dataset represents the largest compilation of *Vibrio cholerae* genomes from Africa and is a valuable contribution to the field.

(Remarks on code availability)

The authors have updated their github repository to include those changes in the revision process and they have added the log and tree files for discrete phylogeographic reconstruction. In addition, the authors have provided a json file to visualize their dated tree using nextstrain which make it easy to visualize the tree results.

Reviewer #3

(Remarks to the Author)

To the Authors,

Thank you for your comprehensive response and the substantial effort invested in addressing my initial review. The CholGEN consortium represents an important advancement in cholera genomic surveillance across Africa, and the dataset you have assembled will undoubtedly prove valuable for understanding regional transmission patterns. I commend the technical improvements made, including the corrected antimicrobial resistance figure and efforts to improve repository accessibility.

I want to commend your decision to provide the complete BEAST files, maximum likelihood trees, and analysis within your GitHub repository. This level of transparency significantly enhances the reproducibility of your work and represents excellent scientific practice. While thorough examination of these files has revealed some areas for improvement, your commitment to open data sharing strengthens the overall contribution and allows for constructive scientific dialogue.

However, several scientific issues require resolution before the manuscript can be considered suitable for publication.

1. BEAST analyses

Your response states that "All continuous parameters were confirmed to have ESS values greater than 200." However, examination of the provided BEAST log files shows some parameters below this threshold. The phylogenetic dating analysis contains a T15a clade age parameter with an ESS of 159, which while close to acceptable, contradicts your categorical statement.

The discrete trait analysis presents a more substantial concern, with multiple location transition rate parameters showing ESS values well below 200, including several under 100 and one as low as 56. While I understand that achieving convergence across all parameters in complex phylogeographic models can be challenging, these values suggest the MCMC chains may not have run sufficiently long to reliably estimate transmission rates between specific country pairs.

Given that cross-border transmission patterns represent a key finding of your study, ensuring robust statistical support for these inferences would strengthen the manuscript's conclusions. Extended MCMC runs or alternative approaches (such as running BEAST on GPUs) might help achieve the convergence needed for confident interpretation of these results.

2. Phylogenetic interpretation of AFR15 introductions

Your response claims that the "split between Asian and African sequences has posterior support of 100%" and that "sequences from the putative second, smaller introduction do descend from Asian sequences, making a second introduction the most parsimonious explanation." However, careful examination of your phylogenetic analyses suggests these conclusions require substantial qualification.

While the smaller AFR15 clade itself has a well-supported MRCA, its placement within the broader context of Asian genome lineages lacks robust support. Your response references an "MRCA of Asian sequence" with "posterior support of 78%," but I may be examining a different tree or node than intended, as the values I observe are slightly different. In the tree provided, the node serving as MRCA to the two Asian lineages (one containing the smaller African clade) appears to show posterior support of 0.82. While this level of support approaches conventional thresholds, the broader phylogenetic picture suggests caution in interpreting this region.

More concerning, your maximum likelihood tree disagrees with the BEAST topology by placing all African T15 genomes together in a single clade. This ML tree also shows poor bootstrap support throughout the relevant nodes. The conflicting results between analytical methods, combined with weak statistical support in both trees, indicate that this portion of the phylogeny is highly unstable. Visualizing the posterior distribution of trees using tools like DensiTree could help identify where these conflicting phylogenetic signals arise and quantify the uncertainty around key nodes.

Your assertion that discrete state analysis demonstrates with "posterior support of 100%" that the ancestor lacks African ancestry must also be qualified given the convergence failures documented above. The underlying topological uncertainty, compounded by the ESS issues in the discrete trait analysis, suggests these definitive conclusions should be tempered until the methodological concerns are resolved.

3. Data availability correction:

This is a minor point, but your response states that sequences from Rouard et al. (NEJM, 2024) "are still not publicly available." However, these data have been accessible on NCBI/SRA since June 2024 (BioProject PRJEB68309) according to the SRA dates for when the reads were made public. Given that this publication describes cholera strains with the same IncA/C plasmid found in your Ugandan isolates, incorporating these sequences into your analysis or discussing their relevance in the context of your findings would strengthen the manuscript.

The consortium's efforts represent valuable progress in genomic surveillance, and I look forward to seeing the revised manuscript.

(Remarks on code availability)

The authors have made a commendable effort to provide comprehensive code and data through their GitHub repository (<https://github.com/CholGen/RegionalAnalysis-2024>). The repository includes the complete computational pipeline, BEAST XML files, log files, and figure generation scripts, which significantly enhances reproducibility.

Version 1:

Reviewer comments:

Reviewer #3

(Remarks to the Author)

I thank the authors for their comments/corrections and commend them on the CholGEN consortium. I look forward to seeing this published.

(Remarks on code availability)

Response to Reviewers

Referee 2

The authors have satisfactorily addressed my previous concerns. They have expanded the dataset, added analyses, acknowledged sampling biases, and clarified language. While some conclusions remain high-level, the dataset represents the largest compilation of *Vibrio cholerae* genomes from Africa and is a valuable contribution to the field.

We thank the referee for their many helpful comments.

Referee 3

To the Authors,

Thank you for your comprehensive response and the substantial effort invested in addressing my initial review. The CholGEN consortium represents an important advancement in cholera genomic surveillance across Africa, and the dataset you have assembled will undoubtedly prove valuable for understanding regional transmission patterns. I commend the technical improvements made, including the corrected antimicrobial resistance figure and efforts to improve repository accessibility.

I want to commend your decision to provide the complete BEAST files, maximum likelihood trees, and analysis within your GitHub repository. This level of transparency significantly enhances the reproducibility of your work and represents excellent scientific practice. While thorough examination of these files has revealed some areas for improvement, your commitment to open data sharing strengthens the overall contribution and allows for constructive scientific dialogue.

However, several scientific issues require resolution before the manuscript can be considered suitable for publication.

We thank the referee for their many helpful comments and have addressed each of their concerns below.

1. BEAST analyses

Your response states that "All continuous parameters were confirmed to have ESS values greater than 200." However, examination of the provided BEAST log files shows some parameters below this threshold. The phylogenetic dating analysis contains a T15a clade age parameter with an ESS of 159, which while close to acceptable, contradicts your categorical statement.

We apologize for the confusion. By "continuous parameters" we specifically meant model parameters that are sampled as part of the BEAST MCMC run, and not summary statistics such as the age of the T15a and T15b clades. We have updated the methods section to clarify this:

Convergence and mixing of the MCMC chains were assessed with Tracer v1.7.2 and Beastiary v1.8.3. We confirmed that all estimated continuous parameters had effective sample sizes greater than 200 and all key summary statistics had effective sample sizes greater than 150.

The BEAST handbook indicates that while ESS values above 200 are preferred, ESS values above 100 are acceptable (See Drummond, Alexei J., and Remco R. Bouckaert. Bayesian evolutionary analysis with BEAST).

Additionally, while the ESS for the age of T15a is slightly less than the other values in the log file, the distribution is stationary in the MCMC, confirmed by two independent chains, and reasonable given the sequence dates in the tree.

The discrete trait analysis presents a more substantial concern, with multiple location transition rate parameters showing ESS values well below 200, including several under 100 and one as low as 56. While I understand that achieving convergence across all parameters in complex phylogeographic models can be challenging, these values suggest the MCMC chains may not have run sufficiently long to reliably estimate transmission rates between specific country pairs.

Given that cross-border transmission patterns represent a key finding of your study, ensuring robust statistical support for these inferences would strengthen the manuscript's conclusions. Extended MCMC runs or alternative approaches (such as running BEAST on GPUs) might help achieve the convergence needed for confident interpretation of these results.

The presence of low ESS values for certain transition rates is expected and accounted for in our analyses. Our discrete phylogeographic reconstruction estimated a large number of transition rates because it included every African country with at least 20 sequences as a discrete location, but our analysis focuses only on supported (non-zero) transition rates (see table below, which lists the ESS values for all transition rates included in **Figure 3**, all of which are above 200).

LocationA	LocationB	Mean Rate	ESS
Cameroon	Nigeria	2.924	605
Cameroon	Niger	0.790	386
DRC	Zambia	1.652	543
DRC	Tanzania	1.035	643
DRC	Uganda	0.618	706
Ghana	Nigeria	0.503	342
Kenya	Tanzania	3.716	569
Kenya	Uganda	0.880	401
Kenya	Zambia	0.675	326
Malawi	South Africa	2.640	595
Malawi	Zambia	2.046	577
Malawi	Mozambique	2.045	656
Malawi	Tanzania	0.498	260
Malawi	Niger	0.516	214
Mozambique	Zimbabwe	1.258	490
Mozambique	Zambia	0.684	358
Niger	Nigeria	2.089	575
South Africa	Zimbabwe	1.494	500
Tanzania	Uganda	0.921	354
Tanzania	Zambia	0.694	430

Specifically, we used Bayesian Stochastic Search Variable Selection to infer which of these transition rates were non-zero. In brief, we modeled the movement between K discrete locations in terms of a $K \times K$ infinitesimal rate matrix Λ , where Λ_{ij} is the instantaneous rate from location i to j . Given the large number of locations and the sparsity of the diffusion process (i.e., cholera appears to be geographically clustered within Africa; see DiPrete et al. 2025. medRxiv), many transition rates will not be supported by the phylogeny (for example, no branches connect a South African sequence to a Nigerian sequence so the South Africa-Nigeria transition rate will only draw from the prior). To account for this sparsity, we used Bayesian Stochastic Search Variable Selection, which limits the number of rates to those that adequately explain the phylogeographic

diffusion process. This Variable Selection parameterizes the instantaneous rate from location i to j , Λ_{ij} , as the product of a rate coefficient, β_{ij} , and a binary indicator variable, δ_{ij} , indicating whether a rate is to be included in the model (0 means not included, 1 means included). The MCMC explores the joint space of β and δ simultaneously, and therefore the discrete trait analysis log file contains columns for the rate and indicator of each location pair. As transition rates with a zero indicator are not included in the model and therefore do not affect the likelihood of the tree, their values aren't expected to converge in the MCMC and understandably have low ESS values. As well, indicator variables are not continuous parameters and therefore are also not expected to have high ESS values. Only transition rates with a non-zero indicator are included in the model and considered in our analysis.

To clarify that only supported transitions are used in our analysis and conclusions, we have updated the **Figure 3** caption from:

(A) [...] Countries were included as states in the discrete state reconstruction if they had more than 20 cholera genomes in our dataset (see Methods). Only locations with a median number of location transitions greater than 0 are labeled. Countries are ordered west-to-east, north-to-south, and so that neighboring countries are next to each other. (B) Top panel: annual density of location transitions for each location pair with median number of location transitions greater than 0 across the posterior distribution of trees.

to:

(A) [...] Countries were included as states in the discrete state reconstruction if they had more than 20 cholera genomes in our dataset (see Methods). Only location pairs with a supported transition rate in at least 50% of the posterior are labeled. Countries are ordered west-to-east, north-to-south, and so that neighboring countries are next to each other. (B) Top panel: annual density of location transitions for each location pair with a supported transition rate in at least 50% of the posterior distribution of trees.

2. Phylogenetic interpretation of AFR15 introductions

Your response claims that the "split between Asian and African sequences has posterior support of 100%" and that "sequences from the putative second, smaller introduction do descend from Asian sequences, making a second introduction the most parsimonious explanation." However, careful examination of your phylogenetic analyses suggests these conclusions require substantial qualification.

While the smaller AFR15 clade itself has a well-supported MRCA, its placement within the broader context of Asian genome lineages lacks robust support. Your response references an "MRCA of Asian sequence" with "posterior support of 78%," but I may be examining a different tree or node than intended, as the values I observe are slightly different. In the tree provided, the node serving as MRCA to the two Asian lineages (one containing the smaller African clade) appears to show posterior support of 0.82. While this level of support approaches conventional thresholds, the broader phylogenetic picture suggests caution in interpreting this region.

More concerning, your maximum likelihood tree disagrees with the BEAST topology by placing all African T15 genomes together in a single clade. This ML tree also shows poor bootstrap support throughout the relevant nodes.

The conflicting results between analytical methods, combined with weak statistical support in both trees, indicate that this portion of the phylogeny is highly unstable. Visualizing the posterior distribution of trees using tools like DensiTree could help identify where these conflicting phylogenetic signals arise and quantify the uncertainty around key nodes.

Your assertion that discrete state analysis demonstrates with "posterior support of 100%" that the ancestor lacks African ancestry must also be qualified given the convergence failures documented above. The underlying topological uncertainty, compounded by the ESS issues in the discrete trait analysis, suggests these definitive conclusions should be tempered until the methodological concerns are resolved.

We have toned down the language in the Results section to more accurately reflect the uncertainty associated with our conclusion of multiple introductions of T15. Specifically we have changed the following text from:

Despite the likely importance of cross-border transmission, our phylogenetic analysis also suggested that there have been multiple introductions of AFR15 into Africa, rather than a single introduction identified by prior investigations (Figure 2A).

To:

Despite the likely importance of cross-border transmission, our phylogenetic analysis also suggested that the AFR15 lineage may represent multiple introductions into Africa from Asia, rather than a single introduction identified by prior investigations (Figure 2A).

Additionally, we added the following text to the Discussion section, highlighting how our estimation of introductions is affected by sampling:

In particular, transmission involving unsampled or undersampled locations may be underappreciated, and our ability to comment on the directionality or absolute number of international transmissions, both within Africa and from other regions into Africa, is limited. Greater availability of global sequence data would help clarify major transmission events, such as the AFR15 introduction, and regional transmission patterns. Refining our understanding of cholera transmission will require continuous and routine genomic surveillance, both during and between cholera outbreaks, and in all countries with reported cholera infections.

While we agree with the reviewer that we do not have conclusive proof of multiple introductions of T15, our data suggests this is the most likely possibility. The reviewer correctly identifies discrepancies between the maximum-likelihood tree and the BEAST analysis, but we have reason to believe the BEAST analyses are more robust for the reasons below: First, we ran the BEAST analysis for 300 million iterations with multiple independent chains, incorporating priors consistent with previous research, allowing it to more thoroughly explore the tree space. In contrast, our maximum-likelihood tree was generated solely as a starting point for the BEAST analysis. We used IQTree2 with default parameters (except for the substitution model and root), and did not confirm if the default number of subtree pruning and regrafting and nearest-neighbor interchange moves sufficiently explored the tree space. While we did include bootstraps, we do not expect them to be particularly informative in outbreak situations such as T15, as bootstrap pseudo-alignments are unlikely to include the few sites which define short edges characteristic of outbreak sub-trees (see Wertheim et al. 2021. Systematic Biology. and Lemoine et al. 2024 Molecular Biology and Evolution). We certainly agree that differences between these trees require careful consideration and have updated our conclusions to reflect the uncertainty in our introduction estimates, however we believe the results from our BEAST analyses, which support multiple introductions, are likely more correct.

3. Data availability correction:

This is a minor point, but your response states that sequences from Rouard et al. (NEJM, 2024) "are still not publicly available." However, these data have been accessible on NCBI/SRA since June 2024 (BioProject PRJEB68309) according to the SRA dates for when the reads were made public. Given that this

publication describes cholera strains with the same IncA/C plasmid found in your Ugandan isolates, incorporating these sequences into your analysis or discussing their relevance in the context of your findings would strengthen the manuscript.

We thank the reviewer for this information and providing us with the relevant BioProject accession ID. We missed that BioProject accession (PRJEB68309) in the New England Journal of Medicine article but can confirm that all of the sequences from the BioProject are included in our genomic data. We had also updated the text to cite all relevant papers when discussing our IncA/C finding. Specifically, we added the following text:

This plasmid has been found sporadically in historical 7PET isolates (Supp. Fig. 6) and was not found in isolates from Uganda collected in 2018-2019, though it was observed in isolates from the 2018-2019 AFR13 outbreak in Yemen and in three Lebanese isolates that phylogenetically clustered with the 2023 isolates from Uganda. Additionally, the plasmid was observed in European travellers returning from Eastern Africa, and a similar IncA/C plasmid containing a modified antimicrobial resistance repertoire was observed in Zimbabwe in 2018, suggesting that such acquisitions are not uncommon.